# Location of Logistics Distribution Center Based on Improved Bald Eagle Algorithm

**Yanfen Tong [1,2] and Xianbao Cheng [3,4,\*]**

1.  College of Economics and Management, Beibu Gulf University, No. 12, Binhai Avenue, Qinzhou 535011, China; purleaf6@foxmail.com
2.  Beibu Gulf Marine Development Research Center, Beibu Gulf University, No. 12, Binhai Avenue, Qinzhou 535011, China
3.  School of Electronics and Information Engineering, Beibu Gulf University, No. 12, Binhai Avenue, Qinzhou 535011, China
4.  Key Laboratory of Beibu Gulf Offshore Engineering Equipment and Technology, Education Department of Guangxi Zhuang Autonomous Region, Beibu Gulf University, Qinzhou 535011, China
\*   Correspondence: luyu1233@foxmail.com

**Abstract:** The location of a logistics distribution center is a complex combinatorial optimization problem, and it is difficult to achieve the best results by traditional mathematical methods. This paper proposes an improved bald eagle search applied to logistics distribution center location selection for the first time, one which uses the chaotic operator to initialize the population, increases the diversity of populations, and introduces a sine and cosine algorithm in the search stage. It increases the global search ability of the algorithm and the ability to jump further out of the local space. Through test function and location simulation experiments, it is proved that the improved algorithm has obvious advantages over other common algorithms in solution accuracy and convergence speed. It can effectively improve the efficiency of logistics distribution when applied to the location of logistics distribution centers. Finally, the improved bald eagle search algorithm is used to optimize the location model of logistics distribution center. The experimental results show that the improved bald eagle search algorithm has good solving ability in this problem, can be obtained to minimize the distribution cost, save the distribution cost effectively and improve the distribution efficiency. It can further optimize the logistics management system and increase the efficiency of logistics enterprises. Compared with similar algorithms, such as WOA, WCA, PSO, the results are improved.

**Keywords:** distribution center location; bald eagle search algorithm; chaotic algorithm; sine and cosine algorithm

## 1. Introduction

With the continuous development of social economy, e-commerce now accounts for more and more in people's lives, and the logistics industry has also developed rapidly. As an important part of e-commerce, logistics distribution centers play an important role in online sales and offline services. As e-commerce shopping has the characteristics of scattered users, large amount of delivery goods, and high real-time quality of user products. how to make logistics distribution more efficient, reduce logistics distribution costs, and shorten distribution time have become important issues for the logistics industry. The location of distribution centers has become a core factor. The distribution center location model has complex nonlinearity and multiple constraints [1], so that, although the traditional mathematical model has a certain effect on the location of logistics distribution centers, it cannot solve this problem fundamentally and there are still certain difficulties.

The location problem of distribution centers has already been studied. The location problem was first proposed by Weber. The location object of the research was the location of the warehouse, and the goal was to minimize the total transportation distance between

the warehouse and each user [2]. Later, more and more people devoted themselves to the research of site selection. There are two main branches of traditional mathematical methods, qualitative methods and quantitative methods. Qualitative methods mainly include analytic hierarchy process [3], delphi method [4], fuzzy evaluation method [5], etc. Quantitative methods include gravity method [6], factor score method [7], mixed-integer programming method [8], coverage model [9], P-median model method [10], etc. Because the logistics centers location problem involves many variables and constraints, it is a typical NP problem. Although the traditional mathematical methods can be optimized to some extent, it is difficult to find the optimal solution for complex problems.

Many scholars have conducted in-depth research on the location problem of non-convex and nonlinear programming with complex constraints and proposed a variety of methods to solve the problem. Among these, the application of advanced artificial intelligence approaches has achieved good results. Advanced artificial intelligence [11] has been well applied in many fields, and good results have been achieved in online learning [12], transportation [13], vehicle scheduling [14–16], and multi-objective optimization [17]. Swarm intelligence algorithm is a kind of advanced artificial intelligence, that at present, many scholars have introduced to the problem of logistics center location. For example, genetic algorithm [18], particle swarm algorithm [19], flower pollination algorithm [20], wolf swarm algorithm [21], whale algorithm [22], simulated annealing algorithm [23], ant colony algorithm [24], etc. The swarm intelligence algorithm is a bionic algorithm that simulates low-intelligence groups seeking food. It can solve non-linear and multi-extremum engineering mathematical problems better and has unique advantages over traditional mathematical methods in solving discontinuous and non-derivative mathematical models.

Particle swarm optimization (PSO) algorithm is a bionic algorithm simulating the predation of birds in flight. PSO algorithm [19] randomly generates an initial population and gives each particle a random velocity. In the flight process, the flight speed and trajectory of the particles are dynamically adjusted according to their own and peer flight experience. The entire population has the ability to fly to a better search area. At present, PSO and its improved algorithm have been widely used in function optimization, neural network training, and fuzzy system control, and there are many references for logistics center location. The advantage of PSO is simple and easy to implement, and there are not many parameters need to be adjusted. The disadvantage is that the performance is not particularly good on some issues. The coding of network weights and the selection of genetic operators are sometimes more troublesome, which also has a certain effect in the application of logistics center location.

Wolf colony algorithm (WCA) is a bionic intelligent algorithm to simulate the predatory behavior of wolf pack. It simulates the three intelligent behaviors of walking, calling and siege, as well as the generation rules of the head wolf of 'winner is king' and the update mechanism of the wolf pack of 'strong survival'. Reference [21] combined this with an immune algorithm to produce a new hybrid algorithm for logistics center location. This paper did not compare it with other algorithms but compared the improved algorithm with the original wolf swarm algorithm, and the results show that it had a certain effect.

Whale optimization algorithm (WOA) is a new swarm intelligence search algorithm derived from imitating the predation behavior of humpback whales, including contraction bounding, spiral updating and hunting. Reference [22] applied this to the location of distribution network centers and compared the results with existing algorithms. The results show that the improved algorithm is more effective in reducing operation cost and maintaining better voltage distribution.

This paper introduces the bald eagle search algorithm [25] to the logistics center location problem for the first time, improves its performance by improving the bald eagle algorithm and proves the feasibility and effectiveness of the algorithm by case study.

## 2. Related Work

### *2.1. Original Bald Eagle Search Algorithm*

The bald eagle search is a new heuristic algorithm proposed by H. A. Alsattar, in 2020 [25]. The algorithm was designed mainly to find the interval optimal value by simulating the behavior of bald eagle in North America. This algorithm has been proven to have a good global search ability. The algorithm is divided into three stages, the selection stage, the search stage and the subduction stage. In the selection stage, bald eagles will choose the search space according to their own position or following other birds come to the prey attachment. When they start to look for food on the water, they will fly on the water and set off in a specific direction; in the search stage, the bald eagle will search the water surface in the selected search space until they find suitable prey; when the bald eagle finds the prey, it will gradually change the flight altitude, and quickly subduct, plunge into the water and successfully capture the prey. According to these three stages of bald eagle predation, the algorithm is also divided into three stages, and its mathematical model is as detailed below.

#### 2.1.1. Select the Search Space

When the bald eagle is in its space, it randomly selects the search area, and determines the best search position by judging the prey concentration. The position update of the bald eagle at this stage is determined by multiplying the prior information on random search times $\alpha$, its mathematical model can be expressed by the following mathematical expression:

$$P_{i,new} = P_{best} + \alpha \cdot r(P_{mean} - P_i) \tag{1}$$

where, $\alpha$ is the position transformation parameter, whose value range is 1.5~2, $r$ is a random number between 0 and 1, $P_{best}$ is the best search position determined according to the previous search, and $P_{mean}$ is the average distribution position of bald eagle determined according to the previous search. $P_i$ is the location of the $i$th bald eagle.

#### 2.1.2. Search Space Prey

At this stage, the bald eagle will explore the prey in the search space. In order to find the prey more quickly, the bald eagle will search for different directions by spiral flight in the search space, so as to find the best diving position. Its mathematical model can be represented by the following polar coordinate equation:

$$\theta(i) = a \cdot \pi \cdot rand \tag{2}$$

$$r(i) = \theta(i) + R \cdot rand \tag{3}$$

$$xr(i) = r(i) \cdot sin(\theta(i)) \tag{4}$$

$$yr(i) = r(i) \cdot cos(\theta(i)) \tag{5}$$

$$x(i) = \frac{xr(i)}{max(|xr|)} \tag{6}$$

$$y(i) = \frac{yr(i)}{max(|yr|)} \tag{7}$$

where: $\theta(i)$ and $r(i)$ are the polar angle and polar diameter of the helical polar coordinate equation, respectively, $a$ and $R$, a parameter that controls the polar coordinate, $a$ varies from 0 to 0.5, which is used to determine the angle between the point searches in the center point, the value range of $R$ is 0.5 to 2, which is used to determine the number of search cycles; rand is a random number of (0,1), $x(i)$ and $y(i)$ is the position of the $i$th bald eagle in the

current coordinates, and the range of value is $(-1, 1)$. The bald eagle position is updated as follows:

$$P_{i,new} = P_i + x(i) \cdot (P_i - P_{mean}) + y(i) \cdot (P_i - P_{i+1}) \tag{8}$$

$P_{i+1}$ is the next update location for the $i$th bald eagle.

### 2.1.3. Subduction Stage

At this stage, the bald eagle subducts towards the prey from the best position in the search space to capture it. At the same time, other bald eagles in the group move towards the best position and attack the prey. The position update at this stage is expressed in polar coordinates as:

$$\theta(i) = a \cdot \pi \cdot rand, r(i) = \theta(i) \tag{9}$$

$$xr(i) = r(i) \cdot \sinh[\theta(i)]; yr(i) = r(i) \cdot \cosh[\theta(i)] \tag{10}$$

$$x(i) = \frac{xr(i)}{\max(|xr|)}; y(i) = \frac{yr(i)}{\max(|yr|)} \tag{11}$$

The updating formula of bald eagle position in subduction is:

$$\begin{cases} \delta_x = x_1(i) \cdot (P_i - c_1 P_{mean}) \\ \delta_y = y_1(i) \cdot (P_i - c_2 P_{best}) \end{cases} \tag{12}$$

$$P_{i,new} = rand \cdot P_{best} + \delta_x + \delta_y \tag{13}$$

Among these: $c_1$ and $c_2$ are the control parameters of the motion intensity of the bald eagle to the best center position, and their ranges are $(1, 2)$.

### 2.2. Location of Logistics Center

The location of a logistics center [26] is a typical NP problem, and it is a kind of location problem. The location problem has been applied in many important fields such as factory locations [27], warehouse locations [28], emergency centers [29], fire stations [30], railway container center site selection [31] and many other important fields. The quality of location is directly related to economic costs and transportation efficiency. The quality of a logistics center location also determines user experience, transportation costs, and commodity timeliness, which are related to the healthy development of e-commerce.

This paper takes the location of general logistics centers as the research object. The location of general logistics centers refers to the process of finding a certain number of locations to establish distribution centers in the locations that logistics must be served. This is to realize the distribution of goods from a logistics distribution center to terminal nodes. The selection index of distribution center is the minimum product of the quantity of goods and the distance from each node to the distribution center [32]. In order to establish the mathematical model of general logistics distribution center, the following assumptions are made:

(1) There is regard for the size of logistics centers and other economic issues;
(2) The distribution centers must meet the requirements of all locations, i.e., the sum of the product of the distance between the center and each node and the volume of goods is the minimum;
(3) There is no regard to other costs

According to the above assumptions, the mathematical expression of the objective function of the general logistics center location is:

$$\min F = \sum_{i \in I} \sum_{j \in M_i} h_i d_{ij} z_{ij}$$

The constraints are as follows:

$$\sum_{j \in M_i} z_{ij} = 1, i \in I \tag{14}$$

$$z_{ij} \leq e_j, i \leq I, j \leq M_i \tag{15}$$

$$\sum_{j \in M_i} e_j = r \tag{16}$$

$$z_{ij}, e_j \in \{0,1\}, i \in I, j \in M_i \tag{17}$$

$$d_{ij} \leq k \tag{18}$$

Among these, min$F$ is the fitness function, which represents the minimum value of the sum of the product of the quantity of goods and the distance from $j$ selected as the logistics center to its distribution terminal node $i$; $h_i$ represents the volume of goods for the $i$th logistics node; $d_{ij}$ denotes the distance between the $i$ position and the nearest logistics center $j$. $z_{ij}$ is a variable of 0 or 1. When $z_{ij}$ is 1, it means that the logistics distribution center j will supply goods for node i, and when $z_{ij}$ is 0, it means that $j$ does not supply goods for node $i$. $e_j$ indicates the demand distribution relationship between the terminal node and the logistics center, and $e_j$ is also a variable of 0 or 1. When $e_j$ is 0, it means that $j$ is an ordinary terminal node. When $e_j$ is 1, it means that $j$ is selected as a logistics center; $r$ represents the number of logistics centers in all current nodes; $k$ is the maximum distance from the selected logistics center $j$ to the node $i$ delivered by it; $M_i$ represents the set of candidate logistics centers whose distances from other node locations to the logistics center are less than $k$; and $I = \{1, 2, \cdots, m\}$ is a collection of all position nodes. Equation (14) means that each node location can only have one logistics center to deliver goods. Equation (15) indicates that each node location can only be delivered by the distribution center and cannot receive goods from other nodes. Equation (16) represents the total amount of selected logistics centers is $r$; Equation (17) represents $z_{ij}$ and $e_j$ are variable of 0 or 1. whose sum $e_j$ is 0 or 1. Equation (18) indicates that the logistics center can deliver goods to all location nodes.

## 3. Algorithm Improvements

### 3.1. Chaos Map Initialization

Like other swarm intelligence algorithms, the bald eagle algorithm randomly distributes the individual positions at the time of initialization. This randomly assigned individual position usually cannot occupy the entire search space, which hinders the diversity of the population. The initial population is evenly distributed in the search space, which is of great help in improving the optimization of the algorithm. Chaotic sequence [33] has randomness, ergodicity and regularity. The basic idea is that chaotic sequence is generated in the interval [0, 1] by mapping relationship, and is then transformed into the individual search space as shown in Figure 1. It can be seen from the graph that the individual is evenly distributed in the whole search space within the dimension of 1000 by chaotic mapping, and basically conforms to the normal distribution law, which greatly increases the diversity of population and provides great help in improving the optimization performance of the algorithm.

There are many models to generate chaotic sequences. In this paper, the chaotic sequences generated by logistics map are used to initialize the bald eagle population. The mathematical expression of logistics mapping is:

$$x^{\mu+1} = \delta x^{\mu}(1 - x^{\mu}), x \in [0, 1]$$

where $\delta$ is the chaotic parameter, $x^{\mu}$ is the chaotic variable, $\delta \in [0, 4]$ the larger the $\delta$, the better the chaos, the value of $\delta$ in this paper is 4.

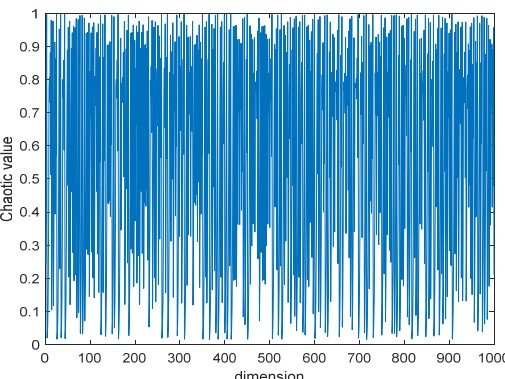

**Figure 1.** Chaos map.

*3.2. Improvements to the Search Phase*

According to the individual update formula of the bald eagle algorithm, the update of the individual position mainly depends on the information of the current position and the average position of the population. The lack of communication between individuals and the low utilization rate of information lead to an inaccurate position update and slow convergence speed of the algorithm. In order to make more communication opportunities between groups to further optimize the exploration and development ability of the bald eagle algorithm, this paper introduces the sine and cosine algorithm [34] as a local optimization operator and embeds it into the bald eagle algorithm. In the later stage of individual position updating, sine cosine operation is used for all bald eagle individuals to guide the updating of bald eagle individuals. The updated formula is as follows:

$$X_i^{t+1} = \begin{cases} X_i^t + r_1 \times \sin(r_2) \times \left| r_3 F_i^t - X_i^t \right| & r_4 < 0.5 \\ X_i^t + r_1 \times \sin(r_2) \times \left| r_3 F_i^t - X_i^t \right| & r_4 \geq 0.5 \end{cases}$$

where $X_i^{t+1} = \left( x_1^i, x_2^i, \cdots, x_d^i \right)^T$ represent the location of individual $i$ in d-dimensional space; $F_i^t = (F_1, F_2, \cdots F_d)^T$ denotes the location of the optimal individual for each generation; $r_1 = a - t \times \frac{a}{T_{max}}$ guides the next generation position region of the $i$th individual; $a$ is a constant, $a = 2$; $r_2$ is a random number between $[0, 2\pi]$, which determines the moving distance that should move towards or away from the target. The flowchart of the improved algorithm is shown in Figure 2:

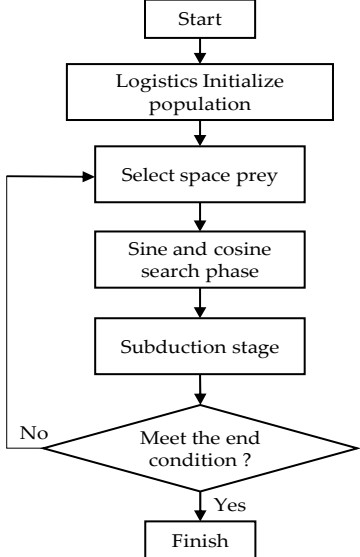

**Figure 2.** Improved algorithm flowchart.

On the one hand, the embedding of the sine-cosine optimization strategy can increase the information exchange between individuals in the bald eagle algorithm. Regardless of the sine or cosine mechanism, bald eagle individuals can communicate with the optimal position in the group and promote the transmission of optimal information in the population, so that each individual can make good use of its own and optimal position information to move to the optimal solution. On the other hand, the sine mechanism can find the optimal solution in the global search, reduce the limitation of the cosine mechanism, and reduce the possibility of individuals falling into the local optimum, while the cosine mechanism can strengthen the local search ability and make up for the slow convergence speed of the global search of the sine mechanism.

## 4. Experiments and Analysis

### 4.1. Test Algorithm Description

In order to verify the performance of the improved algorithm, this paper compares the original bald eagle search [25], PSO algorithm [19], WCA algorithm [21], WOA algorithm [22] and the improved bald eagle search LSCBES algorithm proposed in this paper.

### 4.2. Test Functions

In order to comprehensively test the improved algorithm, this paper selects ten test functions to test the algorithm and compares it with several other intelligent algorithms. Among the selected test functions, $f_1$ is a spherical function. In addition to the global minimum, there are d (dimension) local minimums. It is continuous, concave and unimodal, and $f_2$ has only one optimal value without local optimum. These two functions are more suitable for measuring the convergence accuracy of the algorithm and comparing the convergence speed of the algorithm. $f_3$ is considered a more classical test function. The independent variables of the function are epistatic, so the gradient direction does not change along the axis direction, and the optimization, which is used to test the overall optimization ability of the algorithm, is difficult. $f_4$ is a cone function with single peak and only one global optimum, which is used to test the optimization accuracy of the algorithm. The global minimum of $f_5$ is located in a narrow parabolic valley. Although the valley is easy to find, it is difficult to converge to the minimum, which is mainly used to test the convergence accuracy of the algorithm. When $f_6$ approaches infinite in the definition domain, there will be different jump phenomena at a given interval, and a large number of local extremums will be generated between each step, each of which have high optimization difficulties and are used to test the optimization ability of the algorithm. $f_7$ is a multi-dimensional multi-peak flat-bottom function with random interference. It mainly tests the anti-interference ability of the algorithm and the robustness of the algorithm. $f_8$ has a global minimum point, far from another local optimum, so it is difficult to jump out if it falls into local optimum, and the ability of the algorithm to jump out of local optimum can be tested. The $f_9$ function is a complex nonlinear multi-modal function, and the wave crest presents a high and low fluctuation jump. The solution space has a large number of local minima, which is mainly used to test the population diversity of the algorithm. The $f_{10}$ function is a continuous experimental function modulated by cosine wave. It has many local extremums around the narrow global extremum point, which is mainly used to test the convergence rate of the algorithm.

### 4.3. Test Environment Setting

The parameters settings of the algorithm test are set as follows: the initial population size is 50, the number of iterations is set to 500, and each algorithm is run 30 times. The other parameter settings are consistent with the settings in the literature cited. The experiment was carried out under the 64-bit Windows 10 operating system. The CPU was AMD A8 PRO-7600 B R7, 10 Compute Cores 4C + 6G, 3.10 GHz, and the memory was 8G. Matlab 2018a was used to evaluate the optimization effect by fitness average and standard deviation. The 10 function expressions involved in the test are shown in Table 1.

**Table 1.** Test function expressions.

| Serial Number | Function | Dimension | Search Range | Minimum |
|---|---|---|---|---|
| 1 | $f_1(x) = \sum\limits_{i=1}^{n} x_i^2$ | 50 | $[-100, 100]$ | 0 |
| 2 | $f_2(x) = \sum\limits_{i=1}^{n} |x_i| + \prod\limits_{i=1}^{n} |x_i|$ | 50 | $[-10, 10]$ | 0 |
| 3 | $f_3(x) = \sum\limits_{i=1}^{30} \left( \sum\limits_{j=1}^{i} x_i \right)^2$ | 50 | $[-100, 100]$ | 0 |
| 4 | $f_4(x) = \max\limits_{i}\{|x_i|, 1 \le i \le 30\}$ | 50 | $[-100, 100]$ | 0 |
| 5 | $f_5(x) = \sum\limits_{i=1}^{n-1} [100(x_{i+1} - x_i^2)^2 + (x_i - 1)^2]$ | 50 | $[-30, 30]$ | 0 |
| 6 | $f_6(x) = \sum\limits_{i=1}^{n} (|x_i + 0.5|)^2$ | 50 | $[-100, 100]$ | 0 |
| 7 | $f_7(x) = \sum\limits_{i=1}^{n} ix_i^4 + random[0,1)$ | 50 | $[-1.28, 1.28]$ | 0 |
| 8 | $f_8(x) = \sum\limits_{i=1}^{n} \left( -x_i \sin\left( \sqrt{|x_i|} \right) \right)$ | 50 | $[-500, 500]$ | $-12{,}659.5$ |
| 9 | $f_9(x) = \sum\limits_{i=1}^{n} \left[ x_i^2 - 10\cos(2\pi x_i) + 10 \right]^2$ | 50 | $[-5.12, 5.12]$ | 0 |
| 10 | $f_{10}(x) = -20\exp(-0.2\sqrt{\frac{1}{n}\sum\limits_{i=1}^{n} x_i^2} - \exp(\frac{1}{n}\sum\limits_{i=1}^{n}\cos(2\pi x_i)) + e + 20$ | 50 | $[-32, 32]$ | 0 |

The optimization results of each algorithm are shown in Table 2:

**Table 2.** Running results.

| Test Function | Index | PSO | WCA | WOA | BES | LSCBES |
|---|---|---|---|---|---|---|
| $f_1$ | Average Value | $1.78 \times 10^1$ | $7.65 \times 10^{-11}$ | $3.46 \times 10^{-14}$ | $2.35 \times 10^{-36}$ | 0 |
| | Standard Deviation | $3.26 \times 10^0$ | $4.37 \times 10^{-11}$ | $2.15 \times 10^{-14}$ | $1.27 \times 10^{-36}$ | 0 |
| $f_2$ | Average Value | $2.33 \times 10^1$ | $6.38 \times 10^{-7}$ | $9.26 \times 10^{-13}$ | $4.96 \times 10^{-183}$ | 0 |
| | Standard Deviation | $4.37 \times 10^0$ | $4.96 \times 10^{-7}$ | $4.13 \times 10^{-12}$ | $1.49 \times 10^{-185}$ | 0 |
| $f_3$ | Average Value | $1.16 \times 10^2$ | $2.87 \times 10^1$ | $7.31 \times 10^{-10}$ | $3.67 \times 10^{-223}$ | 0 |
| | Standard Deviation | $4.46 \times 10^0$ | $1.74 \times 10^0$ | $2.33 \times 10^{-10}$ | 0 | 0 |
| $f_4$ | Average Value | $2.45 \times 10^0$ | $5.43 \times 10^{-1}$ | $8.39 \times 10^{-13}$ | $3.21 \times 10^{-226}$ | 0 |
| | Standard Deviation | $7.46 \times 10^1$ | $2.76 \times 10^{-1}$ | $2.91 \times 10^{-13}$ | $1.23 \times 10^{-227}$ | 0 |
| $f_5$ | Average Value | $7.29 \times 10^3$ | $7.77 \times 10^2$ | $2.15 \times 10^1$ | $3.45 \times 10^{-6}$ | $1.43 \times 10^{-6}$ |
| | Standard Deviation | $3.18 \times 10^3$ | $6.45 \times 10^2$ | $1.46 \times 10^1$ | $2.94 \times 10^{-6}$ | $4.36 \times 10^{-6}$ |
| $f_6$ | Average Value | $2.44 \times 10^1$ | $8.14 \times 10^{-10}$ | $6.45 \times 10^{-9}$ | $3.77 \times 10^{-6}$ | $7.92 \times 10^{-22}$ |
| | Standard Deviation | $5.73 \times 10^0$ | $5.37 \times 10^{-10}$ | $2.78 \times 10^{-9}$ | $8.73 \times 10^{-7}$ | $6.45 \times 10^{-22}$ |
| $f_7$ | Average Value | $6.26 \times 10^1$ | $9.46 \times 10^{-3}$ | $7.26 \times 10^{-3}$ | $2.33 \times 10^{-5}$ | $3.92 \times 10^{-6}$ |
| | Standard Deviation | $2.91 \times 10^1$ | $3.42 \times 10^{-2}$ | $5.42 \times 10^{-3}$ | $4.26 \times 10^{-5}$ | $2.35 \times 10^{-6}$ |
| $f_8$ | Average Value | $-7.65 \times 10^3$ | $-5.64 \times 10^3$ | $-2.15 \times 10^3$ | $-2.13 \times 10^3$ | $-7.94 \times 10^2$ |
| | Standard Deviation | $2.45 \times 10^3$ | $4.94 \times 10^2$ | $8.92 \times 10^1$ | $7.96 \times 10^{-3}$ | $4.32 \times 10^2$ |
| $f_9$ | Average Value | $1.66 \times 10^2$ | $7.86 \times 10^{-6}$ | $5.45 \times 10^{-16}$ | $6.57 \times 10^{-98}$ | $2.15 \times 10^{-199}$ |
| | Standard Deviation | $2.74 \times 10^1$ | $3.82 \times 10^{-6}$ | $4.37 \times 10^{-16}$ | $5.42 \times 10^{-99}$ | $4.37 \times 10^{-199}$ |
| $f_{10}$ | Average Value | $1.08 \times 10^1$ | $7.12 \times 10^{-7}$ | $6.64 \times 10^{-12}$ | $8.42 \times 10^{-16}$ | $2.41 \times 10^{-19}$ |
| | Standard Deviation | $2.32 \times 10^0$ | $2.34 \times 10^{-8}$ | $4.33 \times 10^{-12}$ | $2.11 \times 10^{-18}$ | 0 |

In the running results, the evaluation indicators are set as the mean value and the standard deviation, the mean value reflects the convergence speed of the algorithm, and the standard deviation reflects the stability and robustness of the algorithm. It can be seen from the running results in Table 2 that in the unimodal function $f_1 \sim f_5$, the average value of the improved algorithm is better than other algorithms. Among $f_1 \sim f_4$ these functions, the average value and standard deviation of the improved algorithm are both 0.

This shows that the LSCBES optimization results are stable and have high optimization accuracy. In the multimodal function $f_6 \sim f_{10}$, the improved algorithm has also achieved good results, which shows that the improved algorithm can quickly jump out of the local optimum and obtain the global optimum value, which has obvious advantages in both local and global search.

## 5. Application in Location of Logistics Center

In order to verify the correctness and effectiveness of the improved bald eagle search (LSCBES) in the location of logistics centers, this paper selects 31 logistics nodes in a certain place for verification, and sets up six distribution centers, at the same time, several intelligent algorithms mentioned in this paper are compared. The coordinates of each logistics node and the amount of goods are shown in Table 3. The 31 coordinate points in the table are all measured in the same coordinate system in kilometers. All experiments were carried out in the same experimental environment, the operating system was windows 10, the memory was 8G, and the software used was Matlab 2018a.

**Table 3.** Coordinates of logistics nodes and cargo volume.

| Serial Number | Coordinate | Cargo Volume | Serial Number | Coordinate | Cargo Volume |
|---|---|---|---|---|---|
| 1 | (1625, 2413) | 20 | 17 | (4027, 2106) | 90 |
| 2 | (3710, 924) | 90 | 18 | (4135, 2419) | 70 |
| 3 | (4213, 2256) | 90 | 19 | (3864, 2217) | 100 |
| 4 | (3694, 1403) | 60 | 20 | (3655, 2543) | 50 |
| 5 | (3476, 1537) | 70 | 21 | (4122, 2795) | 50 |
| 6 | (3319, 1558) | 70 | 22 | (4257, 2931) | 50 |
| 7 | (3238, 1231) | 40 | 23 | (3429, 1908) | 80 |
| 8 | (2793, 1546) | 90 | 24 | (3507, 2376) | 70 |
| 9 | (2894, 1793) | 90 | 25 | (3451, 2712) | 80 |
| 10 | (3154, 1425) | 70 | 26 | (3275, 3014) | 40 |
| 11 | (2857, 2236) | 60 | 27 | (3167, 3455) | 40 |
| 12 | (2346, 1498) | 40 | 28 | (3345, 3716) | 60 |
| 13 | (2476, 1154) | 40 | 29 | (2296, 2437) | 70 |
| 14 | (1819, 1479) | 40 | 30 | (3004, 3152) | 50 |
| 15 | (1684, 829) | 20 | 31 | (2754, 3666) | 30 |
| 16 | (3729, 1683) | 80 | | | |

The results of several algorithms are compared as shown in Table 4:

**Table 4.** Comparison of the results of several algorithms.

| Algorithm | Site Selection Plan | Distance * Cargo Volume | Number of Iterations |
|---|---|---|---|
| LSCBES | (5, 9, 12, 18, 25, 27) | $6.1069 \times 10^5$ | 33 |
| BES | (3, 5, 9, 12, 20, 27) | $6.1934 \times 10^5$ | 28 |
| WOA | (5, 11, 14, 18, 25, 27) | $6.3114 \times 10^5$ | 30 |
| WCA | (5, 8, 14, 18, 20, 27) | $6.2412 \times 10^5$ | 52 |
| PSO | (4, 6, 12, 18, 25, 27) | $6.4463 \times 10^5$ | 42 |

According to the evaluation index of logistics distribution center selection, the product of the amount of goods and the distance from each node to the distribution center is the

smallest. It can be seen from Table 4 that the location of the LSCBES algorithm is the most reasonable, and the product of the obtained distance and the amount of goods is the smallest. Compared with other algorithms, its result is an order of magnitude smaller, which shows that the improved algorithm proposed in this paper has high optimization accuracy and can find a more reasonable distribution center location. The number of iterations indicates the operational efficiency of the algorithm. From the results in Table 4, it can be seen that BES has the least number of iterations when it reaches the optimal value. LSCBES is affected by adding chaotic mapping and positive chord operator, so its execution efficiency is affected. However, on the issue of logistics center location, since the algorithm finds the optimal value in a small number of iterations, the time difference between WCA with the highest number of iterations and BES with the lowest number of iterations is 0.261s, so the impact of the number of iterations is small and can be ignored.

Figures 3–7 visually show the addressing results of several algorithms:

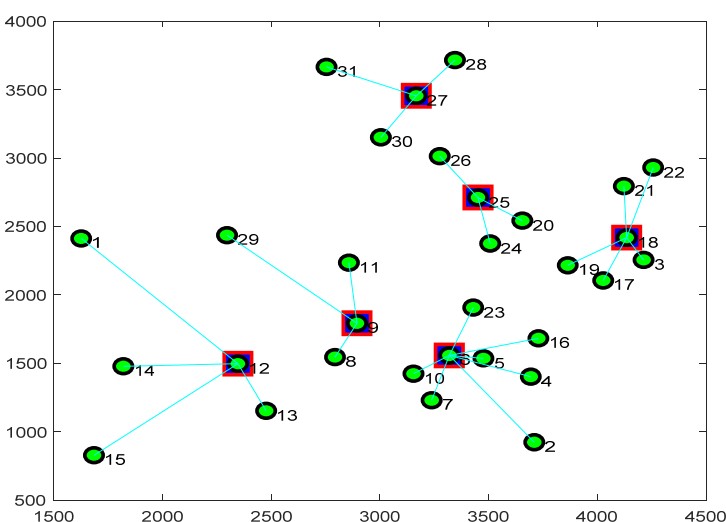

**Figure 3.** Optimization results of LSCBES distribution center.

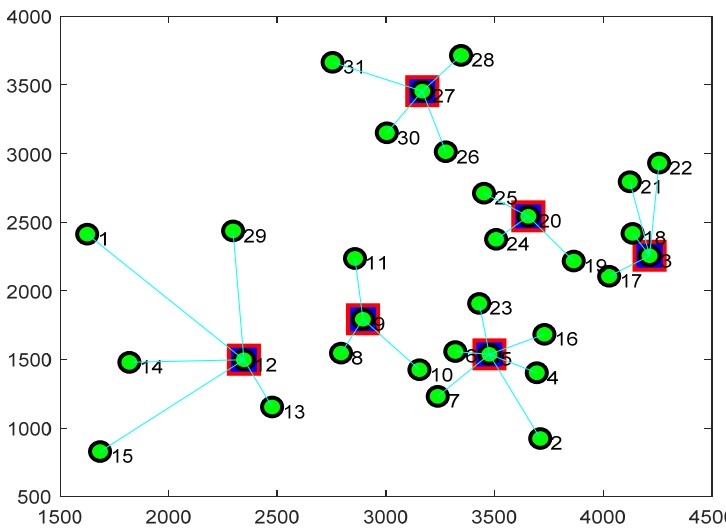

**Figure 4.** BES distribution center optimization results.

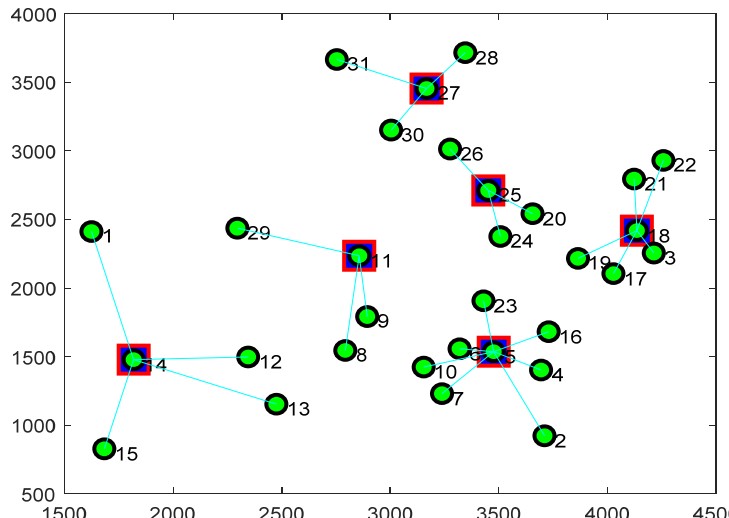

**Figure 5.** WOA distribution center search results.

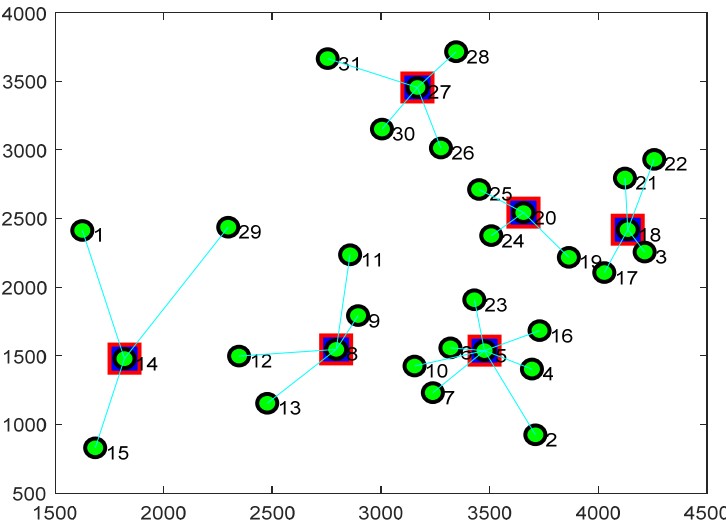

**Figure 6.** WCA distribution center optimization results.

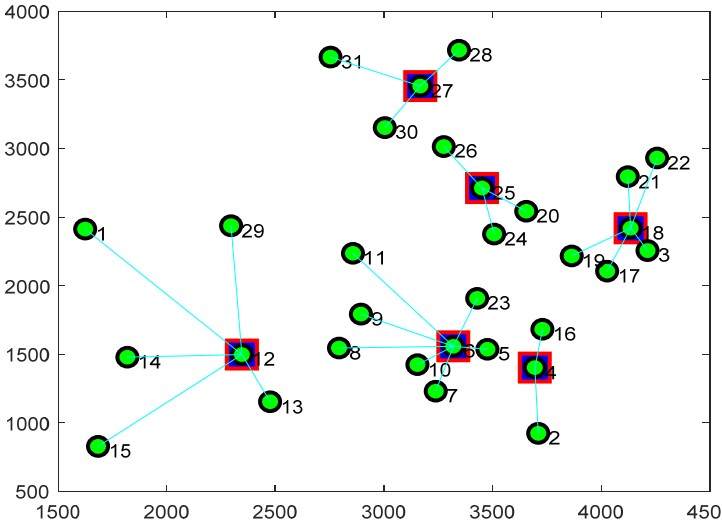

**Figure 7.** PSO distribution center search results.

In Figures 3–7, the circle represents the logistics node, the box represents the selected logistics distribution center, and the number next to the node represents the serial number of the node. From the figures, it can be seen that when using different algorithms for optimization, the location results of logistics distribution center are different, and the scope of distribution by the distribution center is also different. By using Matlab2018a system simulation software to calculate the example, the results show that, according to the evaluation index of [32], the minimum total consumption of the location system of the selected six distribution centers is the optimization result of the improved algorithm in this paper. The distribution centers are 5, 9, 12, 18, 25, 27. The maximum consumption is the result of PSO optimization. The detailed addressing scheme is shown in Table 5:

**Table 5.** Addressing schemes of each algorithm.

| LSCBES | | BES | | WOA | | WCA | | PSO | |
|---|---|---|---|---|---|---|---|---|---|
| Distribution Centera | Distribution Range | Distribution Centera | Distribution Range | Distribution Centera | Distribution Range | Distribution Centera | Distribution Range | Distribution Centera | Distribution Range |
| 5 | 2, 4, 6, 7, 10, 16, 23 | 3 | 17, 18, 21, 22 | 5 | 2, 4, 6, 7, 10, 16, 23 | 5 | 2, 4, 6, 7, 10, 16, 23 | 4 | 2, 16 |
| 9 | 8, 11, 29 | 5 | 2, 4, 6, 7, 16, 23 | 11 | 8, 9, 29 | 8 | 9, 11, 12, 13 | 6 | 5, 7, 8, 9, 11, 23 |
| 12 | 1, 13, 14, 15 | 9 | 8, 10, 11 | 14 | 1, 12, 13, 15 | 14 | 1, 15, 29 | 12 | 1, 13, 14, 15, 29 |
| 18 | 3, 17, 19, 21, 22 | 12 | 1, 13, 14, 15, 29 | 18 | 3, 17, 19, 21, 22 | 18 | 3, 17, 21, 22 | 18 | 3, 17, 19, 21, 22 |
| 25 | 20, 24, 26 | 20 | 19, 24, 25 | 25 | 20, 24, 26 | 20 | 19, 24, 25 | 25 | 20, 24, 26 |
| 27 | 28, 30, 31 | 27 | 26, 30, 31, 28 | 27 | 28, 30, 31 | 27 | 26, 28, 30, 31 | 27 | 28, 30, 31 |

## 6. Conclusions

The location of the logistics distribution center is related to the operating cost and user experience of the e-commerce industry. A reasonable center location can improve the efficiency of logistics and distribution. However, the location problem of logistics distribution center is a kind of complex system optimization problem, which belongs to a kind of combinatorial optimization. It is difficult to solve this kind of problem with traditional mathematical methods. Based on the successful experience of a swarm bionic intelligence algorithm in the existing combinatorial optimization problems, the improved bald eagle algorithm was applied to the location of distribution centers. The experimental results show the effectiveness of this application.

Since the location of distribution center is a relatively diverse and complex research problem, the research on the location of logistics distribution center based on the improved bald eagle search algorithm in this paper has some shortcomings and needs to be further studied. For example, only the location problem of distribution center under certainty factors is considered, but the location problem of distribution center under uncertain factors is not considered; only the distance between logistics nodes and the amount of goods at the nodes are considered, but the load of vehicles is not considered, such as cargo restrictions, time, consumption, etc. Therefore, the location problem of logistics distribution center based on bald eagle search algorithm in this paper still has further research space which needs to be further deepened and expanded.

**Author Contributions:** Conceptualization, Y.T.; data curation, Y.T.; formal analysis, X.C.; investigation, Y.T.; methodology, Y.T.; resources, Y.T.; software, X.C.; validation, X.C.; visualization, Y.T.; writing—original draft, Y.T.; writing—review and editing, X.C. All authors have read and agreed to the published version of the manuscript.

**Funding:** This work is partially supported by the Key research base of Humanities and Social Sciences in Guangxi Universities "Beibu Gulf Ocean Development Research Center", and partially supported by the High-level Personnel Startup Program of Beibu Gulf University (Grant NO. 2018KYQD39).

**Institutional Review Board Statement:** Not applicable.

**Informed Consent Statement:** Not applicable.

**Data Availability Statement:** Not applicable.

**Conflicts of Interest:** The authors declare no conflict of interest.

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
