# Peer review of "Location of Logistics Distribution Center Based on Improved Bald Eagle Algorithm"

_sustainability, doi:10.3390/su14159036_

Round 1
Reviewer 1 Report
Improve the result discussion
1. Improve the literature review in the introduction section and highlight the significance of the paper.
2. Figure 1 is enhanced.
3. Draw a flowchart of the proposed strategy.
4. Clearly describe the hardware and software components in the experimental section.
Reviewer 2 Report
Does this work a pure simulation or has any real implication in a real logistic distribution center? For example, what is the unit of the real physical distance in Fig. 2-6? Is it separated 1-kilometer distance between the center and its branches? Please clarify this unit issue in your manuscript. Thanks.
Reviewer 3 Report
Good Comparison Between PSO, WCA, WOA and BES with 10 Test function
Author Response
Thank you very much
Reviewer 4 Report
1- What is the main innovation of your work compared to previous studies?
2- Authors should present the most important results of their research in the abstract of the article.
3-It is better for the authors to present the method used and the results for each reference in the introduction section.
4- It is better for the authors to provide more complete explanations about the functions used in Table 1.
5- Explain the difference between the results obtained in Figures 2 and 3.
6- Considering the calculation time and the obtained results, it seems that the results obtained from PSO method are more appropriate than the other methods. If this is not the case, the authors should provide a more complete explanation and interpretation.
Round 2
Reviewer 4 Report
The paper can be accepted in the present form.
Author Response
Thank you very much
This manuscript is a resubmission of an earlier submission. The following is a list of the peer review reports and author responses from that submission.